# Influence of the Hypoxia-Activated Prodrug Evofosfamide (TH-302) on Glycolytic Metabolism of Canine Glioma: A Potential Improvement in Cancer Metabolism

**DOI:** 10.3390/cancers15235537

**Published:** 2023-11-22

**Authors:** Hiroki Yamazaki, Seio Onoyama, Shunichi Gotani, Tatsuya Deguchi, Masahiro Tamura, Hiroshi Ohta, Hidetomo Iwano, Hidetaka Nishida, Peter J. Dickinson, Hideo Akiyoshi

**Affiliations:** 1Laboratory of Veterinary Internal Medicine, Companion Animal Internal Medicine, Department of Companion Animal Clinical Sciences, School of Veterinary Medicine, Rakuno Gakuen University, 582-1 Bunkyodai-Midorimachi, Ebetsu 069-0836, Japant-deguchi@rakuno.ac.jp (T.D.); m-tamura@rakuno.ac.jp (M.T.); h-ohta@rakuno.ac.jp (H.O.); 2Laboratory of Veterinary Biochemistry, Department of Veterinary Medicine, School of Veterinary Medicine, Rakuno Gakuen University, 582-1 Bunkyodai-Midorimachi, Ebetsu 069-0836, Japan; h-iwano@rakuno.ac.jp; 3Laboratory of Small Animal Clinics, Veterinary Teaching Hospital, Graduate School of Veterinary Science, Azabu University, 1-17-71 Fuchinobe, Chuo-ku, Sagamihara 52-5201, Japan; h-nishida@azabu-u.ac.jp; 4Department of Surgical and Radiological Sciences, School of Veterinary Medicine, University of California, Davis, CA 95616, USA; pjdickinson@ucdavis.edu; 5Laboratory of Veterinary Surgery, Graduate School of Life and Environmental Sciences, Osaka Metropolitan University, 1-58 Rinku-Oraikita, Izumisano 598-8531, Japan

**Keywords:** HIF-1α, evofosfamide, glycolysis, glioma, dogs

## Abstract

**Simple Summary:**

This study investigated the anti-glycolytic effects of evofosfamide (EVO) on three canine glioma (GL)-derived cell lines with activated hypoxia-inducible factor 1α (HIF-1α). Our clinical data showed that glycolytic activity was correlated with poorer outcomes in dogs with spontaneous GL. Our in vitro studies showed that EVO inhibited glycolytic metabolism by targeting HIF-1α-positive cells under hypoxic culture conditions, resulting in the suppression of cellular ATP production. Our in vivo studies showed that EVO significantly decreased tumor development compared to controls or temozolo-mid in orthotopic murine GL models. A metabolic analysis demonstrated that EVO suppressed glycolytic activity by eliminating HIF-1α-positive cells. Our findings suggest that EVO may improve cancer metabolism and restore the microenvironment for both canine and human GL.

**Abstract:**

The transcription factor hypoxia-inducible factor 1α (HIF-1α) drives metabolic reprogramming in gliomas (GLs) under hypoxic conditions, promoting glycolysis for tumor development. Evofosfamide (EVO) releases a DNA-alkylating agent within hypoxic regions, indicating that it may serve as a hypoxia-targeted therapy. The aim of this study was to investigate the glycolytic metabolism and antitumor effects of EVO in a canine GL model. Our clinical data showed that overall survival was significantly decreased in GL dog patients with higher HIF-1α expression compared to that of those with lower HIF-1α expression, and there was a positive correlation between HIF-1α and pyruvate dehydrogenase kinase 1 (PDK1) expression, suggesting that glycolytic activity under hypoxia conditions may contribute to poor outcomes in canine GL. Our glycolysis assay tests showed that the glycolytic ATP level was higher than the mitochondrial ATP level in three types of canine GL cell lines by activating the HIF-1 signal pathway under hypoxia conditions, resulting in an overall increase in total cellular ATP production. However, treatment with EVO inhibited the glycolytic ATP level in the GL cell lines under hypoxia conditions by targeting HIF-1α-positive cells, leading to decrease in total cellular ATP production. Our in vivo tests showed that EVO significantly reduced tumor development compared to controls and temozolomide in murine GL models. A metabolic analysis demonstrated that EVO effectively suppressed glycolytic metabolism by eliminating HIF-1α-positive cells, suggesting that it may restore metabolism in canine GLs. The evidence presented here supports the favorable preclinical evaluation of EVO as a potential improvement in cancer metabolism.

## 1. Introduction

Gliomas (GLs) are a commonly aggressive primary brain tumor in both human and canine patients [1,2,3]. Canine GLs that arise spontaneously exhibit similar microscopic characteristics to those of human GLs, including their biological behavior, morphology, and the expression pattern of immunohistochemical markers [1,2]. This similarity suggests that the pathological processes are comparable between human and canine GLs [1,2]. Therefore, canine GLs can serve as a suitable animal model for investigating novel therapeutic options for human GLs [2].

Hypoxic stress intimately reinforces glycolysis activity in human GLs [4]. Hypoxia inducible factor 1α (HIF-1α) has a central role in initiating the cascade of GL metabolic reprogramming. Prolyl hydroxylase domain-containing proteins are inactivated under hypoxic conditions, resulting in HIF-1α becoming stable, as it is not hydroxylated. The stabilized HIF-1α translocates to the nucleus, where it binds to HIF-1β to form an active HIF-1 complex that functions as a transcription factor. This activated HIF-1 signaling pathway induces the expression and functionality of glucose transporter type 1 (GLUT1) and pyruvate dehydrogenase kinase 1 (PDK1) [5]. The activated GLUT1 and PDK1 blocks the conversion of pyruvate to acetyl-CoA and increases glycolysis by inducing mitochondrial suppression [4,5,6,7,8]. This process effectively increases the energy supply for tumor development under hypoxia conditions by shunting pyruvate away from oxidative to glycolytic metabolism [4,5,6,7,8]. Therefore, the expression of HIF-1α and PDK1 could serve as glycolysis metabolic markers for the therapeutic target of human GLs [9,10]. Additionally, targeting the glycolysis pathway could potentially have antitumor efficacy by depleting intracellular ATP levels, and several studies have supported the scientific rationale for the effectiveness of this therapeutic approach [6,8,9,10].

Evofosfamide (EVO) (TH-302) is a hypoxia-activated prodrug that releases a brominated version of isophosphoramide mustard (Br-IPM) as a DNA-alkylating compound with cytotoxic properties, specifically targeting the hypoxic areas within tumors. This suggests that it could contribute to a hypoxia-targeting therapy in human GLs [11,12]. We previously reported that EVO can also serve as a novel therapeutic strategy for canine lymphoma [13,14,15]. However, this investigation may be limited, and it remains unknown how EVO affects tumor glycolytic metabolism. Several studies have suggested that EVO suppresses the expression of HIF-1α proteins under hypoxic conditions [16,17,18]. Thus, these suggestions reinforce the prospect of EVO for glycolysis-targeted therapy.

This study aimed to evaluate how EVO affects glycolysis in canine GLs. We hypothesized that EVO could secondarily inhibit the glycolysis pathway in GLs by selectively targeting HIF-1α-positive cells, suggesting that it could potentially contribute to improved metabolism and hypoxic microenvironments in canine GLs.

## 2. Materials and Methods

### 2.1. Medical Record Review and Patient Selection

Medical records from two secondary animal medical facilities (Animal Medical Center at Rakuno Gakuen University and Veterinary Medical Center at Osaka Metropolitan University) were reviewed for dogs with high-grade GLs (oligodendroglioma, astrocytoma, or undefined type) that were diagnosed by histopathology between 2002 and 2022. The inclusion criterion was used to select patients for first-line chemotherapy with temozolomide (TMZ), lomustine (CCNU), nimustine (ACNU), or toceranib phosphate. Dog were excluded if they had other malignant tumors or if they died before the chemotherapy. The clinical information included their age, weight, sex, tumor volume, tumor type, tumor location, clinical signs, overall survival (OS), and follow-up information. The tumor volume was assessed using post-contrast T1-weighted images in the transverse plane. All clinical data were approved by the Institutional Review Board and consent was obtained from the owners.

### 2.2. Immunohistochemical Analyses (IHC)

The levels of HIF-1α and PDK-1 proteins were assessed in paraffin-embedded tumor tissues using immunohistochemical staining (IHC). Dogs with GLs were allocated to two groups: those with high levels of HIF-1α (>median level) and those with low HIF-1α levels (<median level) as previously reported [14]. A statistical analysis was performed to determine the correlation coefficient between the rates of HIF-1α- and PDK1-positive cells. Details of the methods and statistical test are provided in the Appendix A.

### 2.3. Cell Lines and Cultures

Canine GL cell lines (G06A, J3TBg, and SDT3G) were used in our study and were provided by Prof. Peter J. Dickinson. A Mycoplasma Test Kit (Biological Industries, Connecticut, Beit Haemek, Israel) was used to confirm that all cells were free of Mycoplasma contamination. The morphology and growth kinetics of the cells were confirmed to remain consistent throughout the experiment. The cells were cultured using a hydrogel 3-dimensional (3D) culture technique using VitroGel 3D-RGD (The Well Bioscience, North Brunswick, NJ, USA), as previously reported [14]. Briefly, 200 μL of a 1–2 × 10^4^ cell/mL suspension of the cells were mixed with the hydrogel and seeded into 24-well culture plates. Each well also received 250 μL DMEM (Gibco, Grand Island, NY, USA) containing 10% heat-inactivated fetal bovine serum, 1% L-glutamine (Cosmo Bio, Tokyo, Japan), 1% penicillin (Cosmo Bio, Tokyo, Japan), and 1% streptomycin (Cosmo Bio). The plates were then incubated under normoxic (21% O_2_) or hypoxic (1% O_2_) conditions with 5% CO_2_ at 36–38 °C in a multi-gas incubator (APM-30DR, ASTEC Inc., Fukuoka, Japan). The culture medium was regularly changed to provide sufficient nutrition.

### 2.4. Protein Assay

After 24 h of normoxic or hypoxic culture, enzyme-linked immunosorbent assay (ELISA) was used to check for the presence of whole and nuclear HIF-1α proteins, and the relative protein levels were determined by absorbance. Correspondingly, after 7 days of normoxic or hypoxic culture, cells were exposed to 50% inhibitory concentrations (IC50) of TMZ and EVO for 24 h, and whole HIF-1α proteins were determined via ELISA. After 24 h of normoxic or hypoxic culture, Western blotting was used to check for the presence of GLUT1 and PDK1 proteins. Details are provided in the Appendix A.

### 2.5. Reagents

TMZ (Sigma, St Louis, MO, USA) and EVO (TH-302; Threshold Pharmaceuticals, South San Francisco, CA, USA) were prepared for use in treating the cell lines or mice by dissolving and diluting them in either 0.05% dimethyl sulfoxide or PBS, in accordance with the manufacturer’s protocols and previous reports [14,15,19].

### 2.6. Cell Viability and Cytotoxicity Assay

The rates of cell viability were determined after 7 days of normoxic or hypoxic culture. In addition, cell lines were treated with TMZ (0, 5, 50, or 500 μM) or EVO (0, 1, 10, or 100 μM) for 24 h, and the rates of cell viability and apoptosis were assessed. Cell viability and proliferation rate was quantified using Cell Proliferation Kit I (Roche, Indianapolis, IN, USA). Optical densities of the wells were measured at 570 nm using an iMark microplate spectrophotometer (Bio-Rad, Hercules, CA, USA); these data were used to determine the relative rates of cell viability. The rate of apoptosis was measured via flow cytometry using an Annexin V-FITC Apoptosis Detection Kit (Blue Heron Bio-technology, Bothell, WA, USA). Details are provided in the Appendix A.

### 2.7. Measurement of Cellular ATP and Lactate Levels

After 7 days of normoxic or hypoxic culture, intracellular ATP and lactate concentration were assessed using the Glycolysis/OXPHOS Assay Kit (Doujindo, Kumamoto, Japan). Briefly, the cells were seeded into 96-well plates (1.0 × 10^5^ cells/well) and incubated for 5 h in medium containing 1 mM of oligomycin to inhibit mitochondrial oxidative ATP production or 25 mM of 2-deoxy-D-glucose (2DG) to inhibit glycolytic ATP production. After incubation, a luciferase solution was added to the supernatant collected from each well, and the relative light unit (RLU) of each well was measured using the microplate spectrophotometer to determine the relative levels of glycolytic and mitochondrial oxidative ATP production. Similarly, after 7 days of normoxic or hypoxic culture, they were treated for 24 h with IC50 values of TMZ, EVO, and 2DG, and cellular ATP levels were determined.

To measure lactate levels, nicotinamide adenine dinucleotide and glutamate pyruvate transaminase were added to 96-well plates previously described. Then, the enzymatic reaction was started by adding lactate dehydrogenase to each sample and incubating it at 37 °C for 30 min. The OD450 was measured using the microplate spectrophotometer to determine the relative levels of lactate production. Correspondingly, after 7 days of normoxic or hypoxic culture, they were treated for 24 h with IC50 values of TMZ, EVO, and 2DG, and lactate levels were determined.

### 2.8. Experimental Animals

Four-week-old male BALB/cSlc-nu/nu mice (Japan SLC Inc., Shizuoka, Japan) were housed in a controlled room with specific pathogen-free conditions. The room was maintained at a temperature of 24 °C and a humidity range of 40–60%. The mice followed a 12 h light cycle followed by a 12 h dark cycle. Both tap water and regular food (MR-A1; Nosan Corporation, Kanagawa, Japan) were accessible at any time. All mice received a medical check once per day. The experimental protocols were approved by the Institutional Animal Care and Use Committee of Rakuno Gakuen University (approval no. VH22A1). Animal experiments were performed in accordance with the Guidelines for Animal Experimentation of Rakuno Gakuen Unversity.

### 2.9. Murine Xenograft Models

Murine models of canine GL were generated by xenografting G06A, J3TBg, or SDT3G cells as previously reported [20]; the xenografted mice were produced by Sankyo Labo Service Corporation, Inc. (Sapporo, Japan). The cell lines were cultured under hypoxia conditions before being transferred into the mice. Mice were anesthetized with isoflurane and secured on a stereotactic frame. A 5 mm incision was made vertically on the scalp, and a hole with a 0.5 mm diameter was drilled into the calvaria. An aliquot of 1.0 × 10^5^ cells suspended in 10–20 µL Matrigel (BD Biosciences, Tokyo, Japan) was implanted into the drilled hole using a micro-syringe. Two days after implantation, the mice were transported to the experimental animal facility. 

Mice were allocated to four groups (*n* = 5/group): group 1 was the untreated control; group 2 was treated with vehicle; group 3 was treated with TMZ (50 mg/kg, 3 days/week); and group 4 was treated with EVO (50 mg/kg, 3 days/week). The inhibitors were injected intraperitoneally; treatment started on day 1, following a previously reported protocol [15,21]. The general condition of the mice was assessed once a day. All mice were humanely killed 20 days after treatment initiation (on day 21) or if they showed abnormal signs, such as dragging their feet, decreased activity or motion, severe hunchback posture, or a drastic loss of body weight exceeding 20%, following the guidelines on euthanasia at Rakuno Gakuen University. Tumor tissue and blood samples were collected for IHC and blood/biochemical tests at necropsy.

### 2.10. Micro-CT Imaging

The growth of intracranial tumors was monitored using small animal micro-CT (Aloka Latheta LCT-200, Hitachi, Japan) on days 10 and 20 after treatment initiation. All mice were examined in the prone position under anesthesia with isoflurane. The following parameters were used for the CT scans: tube voltage of 50 kV; tube current of 0.5 mA; axial field of view (FOV) of 48 mm for mice; in-plane spatial resolution of 48 µm × 48 µm; and slice thickness of 384 µm for the brain. Brain images were generated by merging four- and eight-signal averages. Total scanning time was approximately 20 min. Measurement of the brain tumor was performed using LaTheta software (version 3.00). Three-dimensional CT images were reconstructed using VGStudio MAX2.0 software (Nihon Visual Science, Tokyo, Japan).

### 2.11. Biochemical Parameters

Blood samples were taken from the mice and centrifuged to isolate serum. Number of neutrophils, aspartate aminotransferase (AST), alanine aminotransferase (ALT), total bilirubin (T-bill), blood urea nitrogen (BUN), and creatinine (Cre) were measured with the Fuji Dri-Chem 700i (FUJIFILM, Tokyo, Japan).

### 2.12. Metabolome Analysis

The metabolomes of three mice that had been xenografted with tumor cells (G06A, J3TBg, or SDT3G) cultured under hypoxic conditions were analyzed from tumor tissue sections as previously reported [15,22,23]. Details are provided in the Appendix A.

### 2.13. Statistical Analysis

Descriptive and comparative statistics were calculated for the entire study population. Numerical data, including age, weight, and tumor volume, were compared using the Wilcoxon rank-sum test, and non-numerical data, including sex and neurological signs, were compared using Fisher’s exact test. The OS of a dog with GL was defined as the duration from the first day of chemotherapy until the day at which all deaths were documented. Right-censored cases were defined as those with incomplete follow-up or unavailable data. Kaplan–Meier survival analysis and the log-rank test were used to compare the OS. Additionally, the study assessed the median OS along with 95% confidence intervals (CI) using Cox proportional hazard models. In vitro and in vivo data were analyzed using one-way analysis of variance (ANOVA). Progression-free survival (PFS) rates for the mice, defined as the duration from treatment initiation until the day when the tumor was visually observed, were compared among the four groups. The levels of HIF-1α protein were compared among cell cultures using the Kruskal–Wallis test. The correlation between levels of HIF-1α and PDK-1 was determined using Pearson’s product-moment correlation test. Metabolome differences were analyzed using Welch’s *t*-test. Quantitative values are presented as the mean ± standard deviation of three separate experiments. All statistical analyses were performed using SPSS v29.0.1 (IBM Corp., Armonk, NY, USA). Statistical significance was set at *p* < 0.05.

## 3. Results

### 3.1. Patient Demographics and Treatment Outcomes

Thirty-six dogs with GLs fulfilled the inclusion criteria. None of the dogs exhibited metastases. Eighteen of the dogs were classified as having high levels of HIF-1α and eighteen had low HIF-1α levels. The characteristics of the two groups of dogs are presented in Table 1. There were no significant differences in the ages, body weights, sex ratios, tumor volumes, or neurological signs between the two groups. A Kaplan–Meier survival curve is presented in Figure 1A. The median OS of the 36 dogs was 104 days (with a range of 24–220 days). The eighteen dogs in the high-HIF-1α category had a median OS of 76 days (with a range of 24–180); this was significantly shorter than that of the median of 118 days (with a range of 36–220) for the eighteen dogs in the low-HIF-1α category (HR: 0.72 (95% CI: 0.69–0.85); *p* = 0.0018). Four of the 36 dogs were censored for loss to follow-up. We used brain tumor samples collected from autopsies for immunohistochemical staining. There was a positive correlation between the rate of HIF-1α- and PDK1-positive cells (Figure 1B; r = 0.868, *p* = 0.00217).

### 3.2. Localization of HIF-1α Protein and *Metabolic Shifts*

The relative levels of whole and nuclear HIF-1α proteins in G06A, J3TBg, and SDT3G cells are shown in Figure 2A. The protein expression was analyzed by absorbance and is expressed as relative intensities (%). In all cells that had been cultured for 24 h under hypoxia conditions, the relative level of whole and nuclear HIF-1α protein was higher than that of normoxia conditions.

The relative levels of whole GLUT1 and PDK1 proteins in G06A, J3TBg, and SDT3G cells are shown in Figure 2B. The immunopositive bands were quantitatively analyzed and expressed as relative intensities (%) against β-actin. Raw data, including the complete blot, is included in the Appendix A. In all cells that had been cultured for 24 h under hypoxia conditions, the level of GLUT1 and PDK1 protein was higher than that under normoxia conditions.

Relative cellular ATP levels were measured after 7 days of normoxic or hypoxic culture, as presented in Figure 2C. The intracellular ATP level in all cells under hypoxia conditions was significantly higher than that in cells under normoxia conditions (G06A, *p* = 0.0042; J3TBg, *p* = 0.0068; and SDT3G, *p* = 0.0098). The rate of glycolytic ATP production in all cells under normoxia conditions was significantly lower than that of mitochondrial ATP production, whereas the rate of glycolytic ATP production under hypoxia conditions was significantly higher than that of mitochondrial ATP production. Lactate production in all cells under hypoxia conditions was significantly higher than that in cells under normoxia conditions (Figure 2D) (G06A, *p* = 0.0092; J3TBg, *p* = 0.0086; and SDT3G, *p* = 0.0075). A comparison of the rates of cell proliferation after culturing for 7 days under normoxia or hypoxia conditions showed a significantly greater rate under hypoxia conditions than normoxia conditions (Figure 2E) (G06, *p* = 0.0092; J3TBg, *p* = 0.0048; and SDT3, *p* = 0.0078).

### 3.3. Cytotoxicity and Glycolysis Suppression

After 7 days of normoxic or hypoxic culture, the sensitivity of the cells to TMZ and EVO is presented in Figure 3A. The viability of G06A, J3TBg, and SDT3G cells treated with 500 μM of TMZ under normoxia and hypoxia conditions was significantly lower than that of the control (vehicle-treated) cells. Similarly, the viability of these cells treated with 10 μM of EVO under hypoxia conditions was significantly lower than that of the control cells. The IC50 values of TMZ in G06A, J3TBg, and SDT3G under normoxia conditions were 180, 260, and 200 μM, respectively, while under hypoxia conditions, they were 720, 500, and 450 μM, respectively. The IC50 values of EVO under normoxia conditions were 160, 360 m and 240 μM, respectively, while under hypoxia conditions, they were 8, 18, and 5 μM, respectively. The sensitivity to TMZ under hypoxia conditions was lower than that under normoxia conditions, while the sensitivity to EVO under hypoxia conditions was higher than that under normoxia conditions.

After 7 days of normoxic or hypoxic culture, the rate of apoptotic cells is presented in Figure 3B. After being treated with the IC50 values of TMZ and EVO for 24 h under normoxia conditions, the apoptosis significantly increased compared to that of the vehicle-treated controls, with TMZ being more effective than EVO. After treatment with TMZ and EVO under hypoxia conditions, the rate of apoptotic cells significantly increased compared to that of the vehicle-treated controls, with EVO being more effective than TMZ.

After 7 days of normoxic or hypoxic culture, the HIF-1α protein expression is presented in Figure 4A. The HIF-1α protein in TMZ-treated and EVO-treated cells under normoxia conditions showed no significant difference compared to that of the vehicle-treated controls. Conversely, the HIF-1α protein in EVO-treated cells under hypoxia conditions significantly decreased compared to that of the controls, while the TMZ-treated cells showed no significant difference.

After 7 days of normoxic or hypoxic culture, the relative cellular ATP amount and lactate production after treatment are presented in Figure 4B,C. Treatment with the IC50 values of TMZ for 24 h and the control (vehicle treatment) resulted in no differences in intracellular ATP levels in all cells under hypoxia conditions compared to those under normoxia conditions, while treatment with EVO and 2DG significantly decreased ATP levels under hypoxia conditions compared to those under normoxia conditions (Figure 4B). Treatment with the IC50 values of TMZ and the control resulted in no differences in lactate production in all cells under hypoxia conditions compared to that under normoxia conditions, while treatment with EVO and 2DG significantly decreased lactate production under hypoxia conditions compared to that under normoxia conditions (Figure 4C).

### 3.4. Antitumor Effects and Adverse Events

Gross tumor volumes were calculated based on CT imaging 10 and 20 days after the treatment (Figure 5A), and the gross tumor weights were measured at autopsy (Figure 5B). The median tumor volumes in the G06A, J3TBg, and SDT3G xenograft mice on 20 days were 748, 916 m and 795 mm^3^ in group 1 (control), 806, 1001, and 703 mm^3^ in group 2 (vehicle), 336, 512, and 474 mm^3^ in group 3 (TMZ), and 160, 258, and 243 mm^3^ in group 4 (EVO), respectively. The tumor volume in groups 3 and 4 significantly decreased compared to that in group 1 (*p* < 0.01), and that of group 4 significantly decreased compared to that in group 3 (*p* < 0.05). The median tumor weights in G06A, J3TBg, and SDT3G xenograft mice were 2.15, 2.82, and 2.34 g in group 1, 2.38, 2.53, and 2.32 g in group 2, 1.22, 1.35, and 1.16 g in group 3, and 0.58, 0.65, and 0.60 g in group 4, respectively. The tumor weight in all murine models was significantly lower in groups 3 and 4 than that in group 1 (*p* < 0.05). 

The Kaplan–Meier curve of the PFS among the four groups is presented in Table 2 and Figure 5C. The PFS for groups 3 and 4 significantly increased compared to that of group 1 (*p* = 0.013 and *p* = 0.007, respectively), and the PFS for group 4 significantly increased compared to that of group 3 (*p* = 0.024). No metastases were detected in the mice at necropsy.

None of the mice showed any neurological or non-neurological signs. There were no significant differences in body weight among the four groups (Figure 6A). The blood and biochemical tests are presented in Figure 6B. No statistically significant differences were observed in BUN and Cre levels among the four groups. Neutrophil counts significantly decreased in groups 3 and 4 compared to those in group 1 (versus group 3, *p* = 0.0018; and versus group 4, *p* = 0.0481), while group 3 showed a significant decrease compared to group 4 (*p* = 0.0349). The AST and T-bill levels significantly increased in group 3 compared to those in group 1 (AST, *p* = 0.00058; and T-bill, *p* = 0.0331), while there were no significant differences between groups 1 and 4. The ALT levels significantly decreased in groups 3 and 4 compared to those of group 1 (versus group 3, *p* = 0.0067; and versus group 4, *p* = 0.0251), while group 3 showed a significant increase compared to group 4 (*p* = 0.0366).

### 3.5. Levels of HIF-1α Protein and Influence on Tumor Metabolism

The levels of HIF-1α protein expression in tumor tissue sections from the four groups are shown in Figure 7A. In three individual murine models, HIF-1α protein levels were significantly lower in group 4 than those in group 1 (G06A; *p* = 0.0062, J3TBg; *p* = 0.0248 and SDT3G; *p* = 0.00784), and there were no significant differences among groups 1, 2, and 3.

Stratified clustering was performed using the detected metabolites in the tumor tissue collected from the mice, and a principal component analysis (PCA) is shown in Figure 7B. The log-fold change in the amount of highly varying metabolites among the four groups was used to perform the PCA. The PCA revealed that group 4 had a high level of variance compared to those of groups 1, 2, and 3. The metabolic pathway clusters were analyzed among the four groups by a heatmap (Figure 7C). Of the eight pathway clusters, the intensity of glycolysis and gluconeogenesis was significantly weaker in group 4 than in that in groups 1, 2, and 3, and there was a similar intensity among the four groups in other pathway clusters. Among all the metabolic pathways, glycolysis and gluconeogenesis exhibited distinctive changes. Of the eleven trace metabolites in glycolysis and gluconeogenesis, nine could be quantified (Figure 7D). Of the nine metabolites, seven metabolites, including Glucose 6-phosphate (G6P), Fructose 6-phosphate (F6P), Dihydroxyacetone phosphate (DHAP), 3-Phosphoglyceric acid (3-PG), Phosphoenolpyruvic acid (PEP), Pyruvic acid, and Lactic acid, showed significantly lower levels in group 4 (EVO) than in group 1, and DHAP showed significantly lower levels in group 3 (TMZ) than in group 1 (Table 3 and Figure 7D).

## 4. Discussion

The study investigated the potential of EVO as a treatment for canine GL, which is a highly malignant brain tumor with a poor prognosis [1,2]. We supported the hypothesis model, as depicted in Figure 8, based on the results of the clinical research. Our strategy focused on the roles of HIF-1α and glycolysis, which may support the progression of canine GL, and the data showed the efficacy of EVO in inhibiting glycolysis processes and suppressing tumor growth both in vitro and in vivo. HIF-1α is a transcription factor that regulates the HIF-1α-related genes involved in glycolysis, angiogenesis, cell survival, and proliferation and is upregulated in human and canine cancers [13,24]. This study first evaluated the association between HIF-1α expression and clinical outcomes in dogs with GLs. Our results showed that higher HIF-1α expression was significantly associated with poorer OS rates compared to those of dogs with lower expression, and there was a strong positive correlation between HIF-1α and PDK1. Although activated HIF-1α has been associated with a poorer prognosis in human GL [4,5,6,7], there are few available data for dogs. 

Our data suggest that HIF-1α-mediated glycolysis activity may result in a poorer outcome for canine GLs. However, this study has limitations due to its small sample size of only 36 dogs and the lack of information regarding the treatment received by the dogs, which may have affected their survival outcomes. Nonetheless, the finding of a similar association between HIF-1α expression and survival in human GLs implies that HIF-1α may have a similar role in GL development in dogs and humans [4,5,6,7].

Our in vitro study investigated the effects of hypoxia on GL cells. These results suggest that the HIF-1α protein is translocated to the nucleus in response to hypoxic stress, and activated HIF-1α promotes glycolysis and increases growth potential due to the enhancement in GLUT1 and PDK1 expression as well as the production of total ATP and lactate in GL cells. This increase in ATP levels suggests that the cells may be relying on alternative metabolic pathways to generate energy in response to hypoxic stress [25]. Glycolysis is a metabolic pathway that converts glucose into pyruvate to provide energy for human GL cells, and it is known as the Warburg effect, a hallmark of cancer cells [25,26,27]. Our findings are consistent with those of previous studies demonstrating the Warburg effect in human glioblastomas [26]. Importantly, our data showed that EVO significantly reduced ATP production and lactate production by selectively targeting HIF-1α-positive cells under hypoxia conditions, and exhibited a higher cytotoxicity and apoptosis than TMZ under hypoxia conditions, suggesting its potential to disrupt the Warburg effect and inhibit glycolysis. A study demonstrated that EVO exhibited antitumor effects in an animal model of human leukemia with elevated glycolytic metabolism [28]. However, this report did not provide complete information on how EVO affects glycolytic metabolism. Our study is the first report of EVO exhibiting the inhibition of the glycolytic pathway by targeting HIF-1α-positive cells under hypoxia conditions.

This in vivo study investigated the therapeutic potential of EVO in murine GL models. The results showed that EVO significantly reduced tumor growth and increased OS compared to TMZ. Interestingly, EVO inhibited glycolysis in tumor cells, which is suggested to be due to the elimination of HIF-1α-positive cells. These findings are consistent with the in vitro data described previously. Several studies suggest that a shift towards glycolytic metabolism can lead to detrimental effects such as chemoresistance or treatment failure in human cancer patients, including those with glioblastomas [29,30]. Moreover, the accumulation of glycolytic metabolites, such as lactate, has been linked to poorer prognoses in human GL patients [31,32]. The inhibition of HIF-1α-positive cells and disruption of glycolysis can be promising mechanisms for clinical benefits in GL patients [33] and may be alternatives to TMZ. EVO is noteworthy for its unique mechanism of cell growth inhibition, which differs from that of TMZ and could result in a higher level of antitumor effects and lower adverse events. Our data demonstrated that EVO reduced bone marrow suppression and hepatotoxicity compared to TMZ. Under hypoxic conditions, EVO induces the production of reactive oxygen species that result in cellular DNA and protein damage [13,14,15]. However, since normal tissues have a minimal presence of hypoxic conditions, EVO remains inactive, and its effects may be limited. Based on our results, EVO potentially improves the metabolism of GL by selectively targeting HIF-1α-positive cells, suggesting its ability to restore the microenvironment of intratumoral hypoxic areas. If this is the case, there are indications of synergistic potential through the combined utilization of TMZ and EVO. EVO demonstrates cytotoxic effects via the HIF-1α mechanism, while TMZ exerts conventional DNA alkylation. The amalgamation of these two therapeutic approaches holds the promise of orchestrating effective attacks through multiple pathways, potentially leading to the improvement of hypoxic microenvironments. Further studies are required to investigate the safety and efficacy of EVO in clinical trials.

This study highlights the importance of metabolic reprogramming in GL progression and response to treatment. The improvement of cancer metabolism by the inhibition of glycolysis has been suggested as a potential strategy for cancer therapy, and our findings reinforce this hypothesis. Moreover, the selective elimination of HIF-1α-positive cells and its downstream effects on tumor metabolism may provide novel insights into the underlying mechanisms of GL treatment development.

## 5. Conclusions

This study provides evidence regarding the antitumor effects of EVO in canine GL models through the improvement of metabolism and suggests its potential as a therapeutic strategy for canine GL patients. Our data indicate a correlation between increased hypoxic regions and worse treatment outcomes in canine gliomas. While the antitumor effects of TMZ decrease against HIF-1α-positive cells in hypoxic areas, EVO may bolster these effects. Therefore, since EVO and TMZ can complement each other’s limitations, their combination could potentially enhance the therapeutic effect. Further studies are required to evaluate the safety and efficacy of EVO in clinical trials and to elucidate the underlying mechanisms of its antitumor effects.

## Figures and Tables

**Figure 1 cancers-15-05537-f001:**
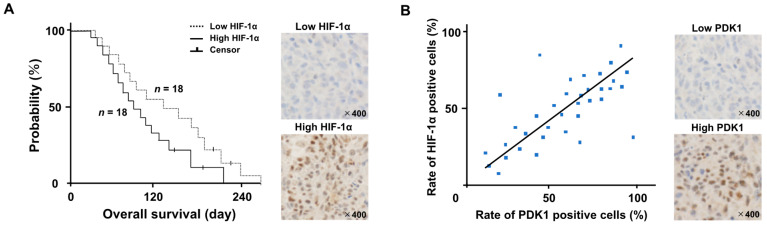
(**A**) Kaplan–Meier survival analysis between the high- and low-HIF-1α groups. The hazard ratio between both groups was 0.72 (95% CI: 0.69–0.85; *p* = 0.0018). (Upper figure; Low HIF-1α image, down figure; High HIF-1α image) (**B**) We used brain tumor samples collected from autopsies for immunohistochemical staining. A Pearson’s correlation scatter plot (blue dot) showed the relationship between the rates of HIF-1α- and PDK1-positive cells. (r = 0.868, *p* = 0.00217). (Upper figure; Low PDK1 image, down figure; High PDK1 image).

**Figure 2 cancers-15-05537-f002:**
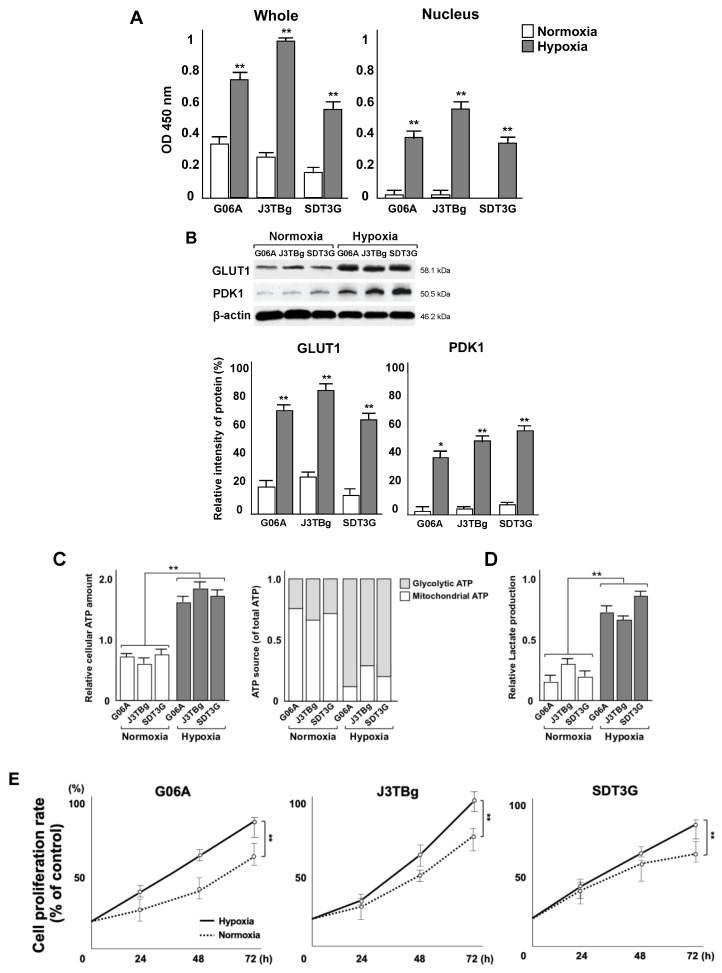
(**A**) Whole and nuclear HIF-1α proteins in G06A, J3TBg, and SDT3G cells after 24 h of normoxic or hypoxic culture. Data are represented as relative value (%). (**B**) Whole GLUT1 and PDK1 proteins in G06A, J3TBg, and SDT3G cells after 24 h of normoxic or hypoxic culture. Immunopositive bands are represented as relative value (%) normalized to β-actin. Raw data, including the complete blot, is included in the Appendix A (**C**) Relative cellular ATP amount (left) and the rate of glycolytic or mitochondrial ATP production (right) after 7 days of normoxic or hypoxic culture. (**D**) Lactate production after 7 days of normoxic or hypoxic culture. (**E**) Cell proliferation rate (%) after 7 days of normoxic or hypoxic culture. * *p* < 0.05, ** *p* < 0.01.

**Figure 3 cancers-15-05537-f003:**
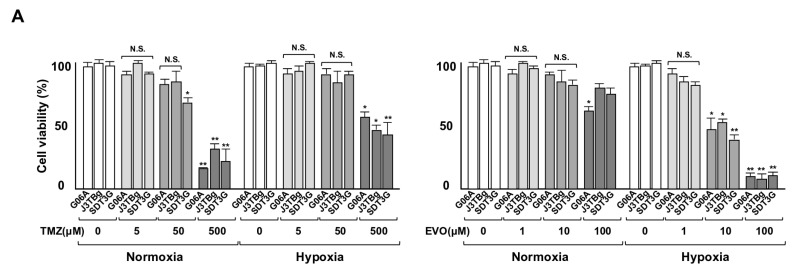
(**A**) Sensitivity to TMZ (0–500 μL) and EVO (0–100 μL) after 7 days of normoxic or hypoxic culture. The cell viability is expressed as a percentage (%) of the control (vehicle-treated) GL cells. (**B**) After 7 days of normoxic or hypoxic culture, the rate (%) of apoptosis in the cell lines after treatments with 50% inhibitory concentration (IC50) values of TMZ and EVO for 24 h. The IC50 values of TMZ in G06A, J3TBg, and SDT3G under normoxia conditions were 180, 260, and 200 μM, respectively, while under hypoxia conditions, they were 720, 500, and 450 μM, respectively. The IC50 values of EVO under normoxia conditions were 160, 360 and 240 μM, respectively, while under hypoxia conditions, they were 8, 18, and 5 μM, respectively. * *p* < 0.05; ** *p* < 0.01; NS, not significant.

**Figure 4 cancers-15-05537-f004:**
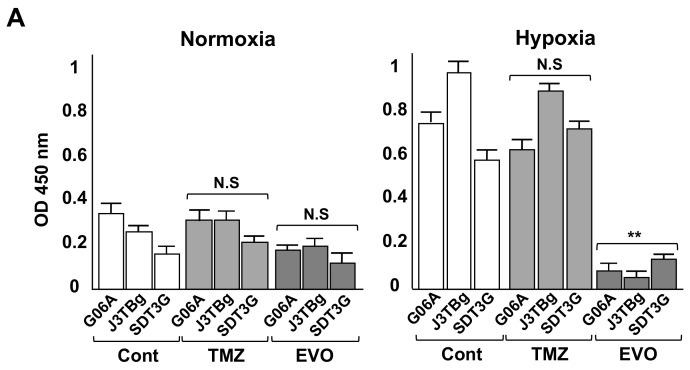
(**A**) After 7 days of normoxic or hypoxic culture, whole HIF-1α protein levels in the cell lines were determined via ELISA following 24 h treatments with the IC50 values of TMZ and EVO. (**B**) After 7 days of normoxic or hypoxic culture, relative cellular ATP amount in the cell lines after treatments with IC50 values of TMZ, EVO, and 2-deoxy-D-glucose (2DG) for 24 h. (**C**) After 7 days of normoxic or hypoxic culture, lactate production in the cell lines after treatments with IC50 values of TMZ, EVO, and 2DG for 24 h. The IC50 values of TMZ in G06A, J3TBg, and SDT3G under normoxia were 180, 260, and 200 μM, respectively, while under hypoxia conditions, they were 720, 500, and 450 μM, respectively. The IC50 values of EVO under normoxia were 160, 360, and 240 μM, respectively, while under hypoxia conditions, they were 8, 18, and 5 μM, respectively. The 2DG was treated with a concentration of 15 mM for all cases. * *p* < 0.05; ** *p* < 0.01; NS, not significant.

**Figure 5 cancers-15-05537-f005:**
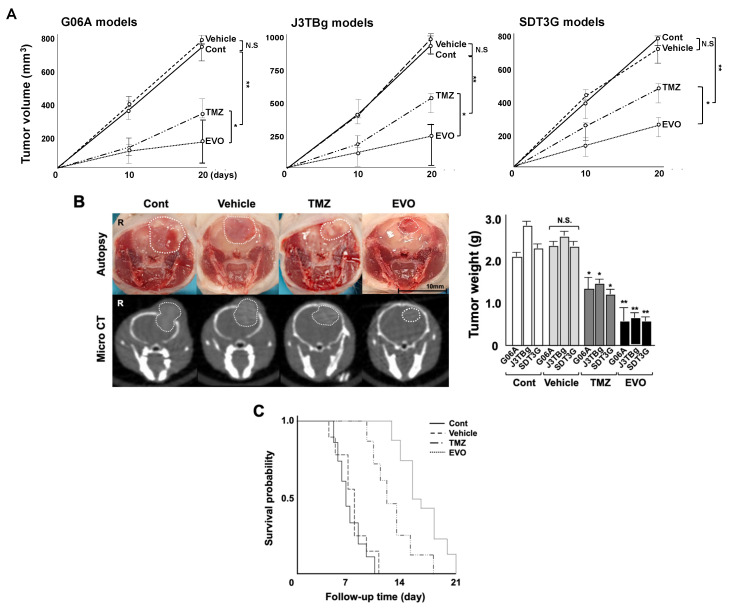
(**A**) Antitumor effects of TMZ and EVO among the four murine groups. Tumor volumes were determined by small animal micro-CT (length × width^2^ × 0.5) 10 and 20 days after the treatments. (**B**) Gross appearance (at autopsy) and micro-CT findings in murine models 20 days after the treatments (The white circle indicated the area of the tumor). Tumor weight was measured after sampling. (**C**) Kaplan–Meier survival analysis among four murine groups. * *p* < 0.05; ** *p* < 0.01; NS, not significant.

**Figure 6 cancers-15-05537-f006:**
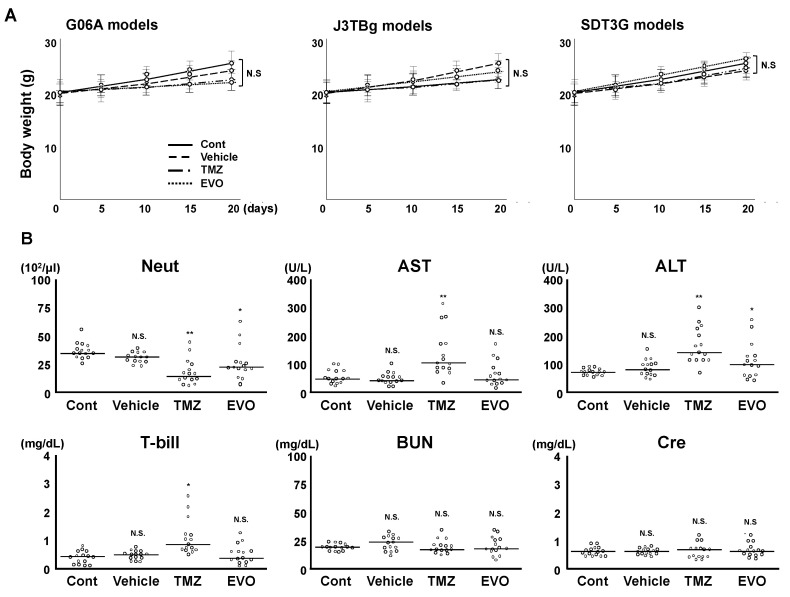
(**A**) Change in bogy weight among four groups. (**B**) Comparative analysis of blood/biochemistry tests among the four murine groups. * *p* < 0.05; ** *p* < 0.01; NS, not significant.

**Figure 7 cancers-15-05537-f007:**
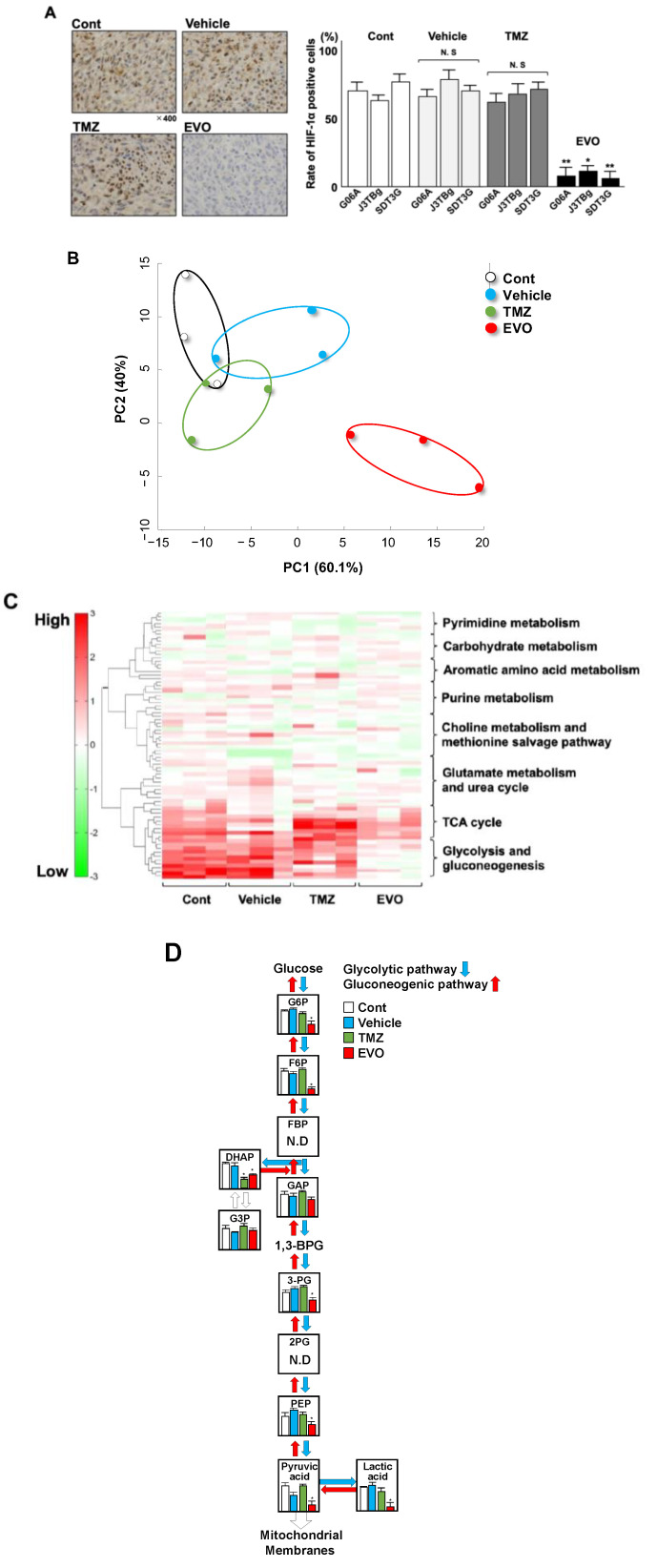
(**A**) Expression of HIF-1α protein in tumors among four murine groups. The expression of the HIF-1α protein was determined with immunohistochemical staining. (**B**) Principal component analysis (PCA) among four murine groups. (**C**) Cluster analysis of 8 metabolic pathways in the heatmap of the four main clusters. Four main clusters included control, vehicle, TMZ, and EVO. (**D**) Comparison of 11 metabolites in glycolysis and gluconeogenesis in the four main clusters. * *p* < 0.05; ** *p* < 0.01. G6P(Glucose 6-phosphate); F6P(Fructose 6-phosphate); FBP(Fructose 1,6-bisphosphate); DHAP (Dihydroxyacetone phosphate); G3P(Glyceraldehyde-3-phosphate dehydrogenase); GAP(Glyceraldehyde-3-phosphate); 3-PG(3-Phosphoglyceric acid); 2PG(2-Phosphoglyceric acid); and PEP(Phosphoenolpyruvic acid).

**Figure 8 cancers-15-05537-f008:**
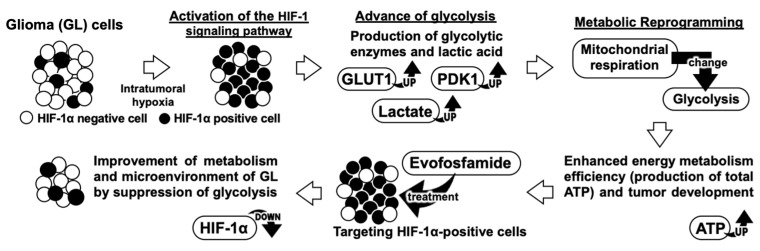
Our research’s hypothesis model.

**Table 1 cancers-15-05537-t001:** Variations in demographic characteristics between the two groups based on their levels of HIF-1α expression.

Variables		High HIF-1α (*n* = 18)	Low HIF-1α (*n* = 18)	*p* Value
Age (year): median (range)		10.6 (4.3–14.5)	11.4 (6.4–16.5)	0.774
Weight (kg): median (range)		9.8 (2.4–42.3)	8.3 (3.1–38.6)	0.531
Sex (*n*)	Male	10 (56%)	11 (61%)	0.469
	Female	8 (44%)	7 (39%)	0.482
Tumor volume (cm^3^): median (range)		1.56 (0.89–3.74)	1.48 (0.82–3.86)	0.085
Tumor type		Oligodendroglioma (14)	Oligodendroglioma (15)	0.362
		Astrocytoma (1)	Astrocytoma (1)	
		Undefined (3)	Undefined (2)	
Tumor location		Hemispheric (15)	Hemispheric (14)	0.545
		Diencephalon (2)	Diencephalon (3)	
		Infratentorial (1)	Infratentorial (1)	
		Parietal (6)	Parietal (5)	0.249
		Temporal (5)	Temporal (7)	
		Front-olfactory (7)	Front-olfactory (6)	
Neurological sign (*n*)		12 (67%)	7 (39%)	0.078

**Table 2 cancers-15-05537-t002:** Comparison of the progression-free survival among the four groups in murine models.

Group	OS; Median Days (Range)	Hazard Ratio	95%CI	*p*-Value
Group 1 (Control)	7.3 (6–10)	1	-	-
Group 2 (Vehicle)	8.1 (5–11)	1.12	0.82–1.36	0.802
Group 3 (TMZ)	11.8 (10–18)	0.78	0.60–0.96	0.013
Group 4 (EVO)	16.0 (13–21)	0.64	0.58–0.85	0.007
Group 3 (TMZ)	-	1	-	-
Group 4 (EVO)	-	0.82	0.73–0.92	0.024

**Table 3 cancers-15-05537-t003:** Comparative analysis of metabolites related to glycolysis and gluconeogenesis.

Pathway Cluster	Compound Name	Comparative Analysis (Ratio)		
		Vehicle * vs. Control	TMZ vs. Control	EVO vs. Control
Glycolysis and gluconeogenesis	Glucose 6-phosphate (G6P)	0.81	0.45	8 × 10^−3^ *
	Fructose 6-phosphate (F6P)	0.71	0.78	3 × 10^−3^ *
	Dihydroxyacetone phosphate (DHAP)	0.84	2 × 10^−2^ *	3 × 10^−2^ *
	Glyceraldehyde-3-phosphate dehydrogenase (G3P)	0.65	0.56	0.12
	Glyceraldehyde-3-phosphate (GAP)	0.34	0.67	0.21
	3-Phosphoglyceric acid (3-PG)	0.54	0.48	4 × 10^−3^ *
	Phosphoenolpyruvic acid (PEP)	0.38	0.21	6 × 10^−3^ *
	Pyruvic acid	0.45	0.08	2 × 10^−3^ *
	Lactic acid	0.39	0.55	3 × 10^−3^ *

* *p* < 0.05; significant difference.

## Data Availability

The data in this study are available on request from the corresponding author.

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
