# Peer review of "Influence of the Hypoxia-Activated Prodrug Evofosfamide (TH-302) on Glycolytic Metabolism of Canine Glioma: A Potential Improvement in Cancer Metabolism"

_cancers, 2023, doi:10.3390/cancers15235537_

Round 1
Reviewer 1 Report (New Reviewer)
Comments and Suggestions for Authors
The authors present an interesting study supporting the utility of evofosfamide (EVO) for the treatment of canine glioma. The findings are of value, however several improvements must be made prior to publication, as detailed below:
MAJOR CONCERNS
- Line 254: The authors need to justify what is the rationale for examining the levels HIF-1alpha in different subcellular compartments (Figure 3A) at the beginning of the paragraph. What is the relevance of HIF-1alpha compartimentalization in relation to its function? Relecvant literature should be cited. Also, why are the levels unchanged in the cytosol? Could it be due to the time point at which the analysis was conducted? Would the levels of HIF-1alpha be elevated in the cytosol as well, when performing the analysis at a longer or shorter time point? Finally, if there is no relevance to the study's main point in looking at subcellular protein levels, the result may be removed.
- lines 269-273 and Figure 3E: why are the cells more viable under hypoxia (stress) than under normoxia? What is causing cell death (up to 50%) under normoxia? This result seems hard to justify. Can the authors provide context?
- Figure 4A displays cell viability upon temozolomide and EVO treatment in cells that had been cultured for 7 days in normoxia or under hypoxia. However, the normoxic cells should be 50% dead according to the results from figure 3E, which would make them inadequate for use in further assays. Can the authors provide additional evidence justifying their experimental design? Microscopic pictures of the cells after 7 days under normoxia or under hypoxia may provide justification on the suitability of these cell cultures for the experiments shown in Figure 4A. Furthermore, the authors should also show in a supplementary figure the relative cell viability of the particular normoxic and hypoxic cultures used for treatment with temozolomide and EVO.
- in figure 8D, the authors report contractiding results for NAD and NADH levels. That is, the levels of each of the two metabolites are different in the reaction from glyceraldehyde 3-phosphate (GAP) to 3-phosphoglycerate (3-PG), compared to their levels as reported in connection with the reaction from pyruvic acid to lactic acid. However, this should not be possible. Can the authors explain? Also, please include an excel file with full results for the metabolomics experiment, including the results for individual replicates, as supplementary.
- on line 467-472 (Conclusions), the authors should clarify the study conclusions. The study results argue that EVO is more effective at targeting hypoxic cells, while temozolomide is more effective at targeting normoxic cells. This could be stated as a conclusion. What are the implications for treatment? Can the authors show the effects on cell viability when combining EVO and temozolomide on a co-culture of normoxic and hypoxic cells, compared to no treatment, or treatment with each compound alone?
MINOR CONCERNS
- move to the discussion section Figure 2 and the statement "We supported the 244 hypothesis model, as depicted in Figure 2, based on the results of the clinical research." (on line 244-245). This figure does not present results but only the working model, therefore should not be part of the Results section.
- specify in the legend of figure 1B which samples are being analyzed (are this the dog's postmortem brain tumor samples?). Similarly, also specify in the text on lines 243-244 which samples are being analyzed: "There was a positive correlation between the levels of HIF-1α and PDK1 protein." (expressed where?)
- can the authors discuss the relevance of the changes in NAD levels shown in figure 8D in relation to the mice's response to therapy? Please also cite relevant literature that discusses how alterations in NAD metabolism affect therapy resistance in glioma, as detailed below:
"Proteomic analysis reveals microvesicles containing NAMPT as mediators of radioresistance in glioma", Panizza et al, Life Science Alliance 2023
- in figure 3E, why is the panel to the right (G06A) a repeat of the panel on the left (also G06A)? what do "g. 2", "B", and "2" refer to?
- how were the levels of HIF-1alpha quantified in Figure 5A? "OD 450 nm" tipically refers to absorbance, which is a measure of the total protein concentration in a sample, rather than of the levels of any individual protein. Can the authors clarify?
Comments on the Quality of English Language
English language should be revised for grammar, syntax, sentences construction and clarity. I advise obtaining support from a professional English language writer to adequately edit the text.
Author Response
Author's Comments to Reviewer #1:
Thank you for reviewing our work. The responses have been attached as a separate document. We kindly request your review once again.

Reviewer 2 Report (New Reviewer)
Comments and Suggestions for Authors
The authors have hypothesized that EVO can inhibit the glycolysis pathway in GL by selectively targeting HIF-1α positive cells, suggesting that it could potentially contribute to improving metabolism and hypoxic microenvironment in canine GL.
This study suggests that targeting glycolysis can act as a therapeutic strategy for canine GL. Further research is needed to develop effective glycolysis-targeted therapies for this disease.
However, there are some limitations such as the limited sample size and the lack of information regarding the treatment received by the dogs. However, the author’s findings provide a starting point for further research on EVO as a potential treatment for canine GL.
The study is well-designed, and the results are promising. The authors have addressed the limitations of their study and provided suggestions for future research.
This study suggests EVO as a promising new treatment for canine GL but further studies are needed to evaluate the effects of EVO and its antitumor effects.
Comments on the Quality of English LanguageThe manuscript is well-written and organized. however , there are minor errors in grammar.
Author Response
Author's Comments to Reviewer #2:
We would like to express our deepest gratitude for your comments and suggestions that helped to improve the quality of our paper.
Ans. Thank you very much for the meaningful comment. As you pointed out, I have added the limitations of the study (L 415-418).
Round 2
Reviewer 1 Report (New Reviewer)
Comments and Suggestions for Authors
The authors satisfactorily responded to all comments and updated the manuscript accordingly. However, the authors have not provided the full results for the metabolomics experiment, including the results for individual replicates, which had been asked as part of thre first revision. It is fundamental to provide complete original data and this information needs to be included as supplementary before the manuscript can be accepted for publication.
Author Response
Author's Comments to Reviewer #1: We would like to express our deepest gratitude for your comments and suggestions that helped to improve the quality of our paper.
- The authors satisfactorily responded to all comments and updated the manuscript accordingly. However, the authors have not provided the full results for the metabolomics experiment, including the results for individual replicates, which had been asked as part of thre first revision. It is fundamental to provide complete original data and this information needs to be included as supplementary before the manuscript can be accepted for publication.
Ans. I apologize for forgetting the previous question. We will submit a data file regarding coenzymes related to the overall metabolic function measured in our current research. Substrates that could not be detected due to measurement limitations have been removed. We have simplified and organized the data file due to its extensive information content. Furthermore, we are not analyzing all the data files; however, we have specifically examined metabolic functions crucial in tumors. Among the data files, we have highlighted the glycolytic pathway, as significant changes were observed after Evo administration, which is presented as results in the paper.

Round 3
Reviewer 1 Report (New Reviewer)
Comments and Suggestions for Authors
All concerns have been addressed.
This manuscript is a resubmission of an earlier submission. The following is a list of the peer review reports and author responses from that submission.
Round 1
Reviewer 1 Report
Comments and Suggestions for Authors
The manuscript by Yamazaki and co-workers is a concise overview of the thriving research field of glycolysis-targeted therapy. Specifically, they reported the excellent influence of hypoxia-activated prodrug evofosfamide on glycolytic metabolism in canine glioma.
I believe the article should be published practically as it is. However, it would be great if the authors take into consideration the following comments/additions:
1. A brief discussion (perhaps in the conclusion section) on some of the present and future challenges of the field, e.g. the antitumor effects of EVO's metabolic efficacy described and how such feature could affect (pre)clinical applications, and the importance of knowing the different courses of action before, during and after their applications.
2. I suggest adding an outline of the postulated mechanism of evofosfamide action of evofosfamide within the HIF-1α inhibition pathway.
3. In lines 87-88, I suggest adding the full name of the drugs mentioned with abbreviation: TMZ, CCNU and ACNU.
4. In line 127, I suggest adding reference/s about the manufacturer’s protocols for preparation of TMZ and EVO prior to use.
5. Line 215 and following lines, what does the abbreviation OS? overall survival mean? add it to the text.
6. Unify the number eighteen to “18” or to “eighteen” in line 233.
7. In Figure 2, add the letter D to the figure (lactate production results).
8. In Figure 3, I suggest adding the used drug concentrations in the figure caption.
9. In Figure 4, I suggest adding specify the drug concentrations used (IC50s) for each drug in the figure caption.
10. In Figure 7D, I suggest adding in the figure caption the full names of the abbreviations used.
11. Please comment in the discussion, whether an improved therapeutic effect could be achieved by combining TMZ with EVO, to improve a normoxic environment.
Reviewer 2 Report
Comments and Suggestions for Authors
In this study the authors investigate the effects of the hypoxia-activated cytotoxic alkylating agent evosfosfamide (EVO) on three canine glioma (GL)-derived cells lines cultured in vitro under normoxia or hypoxia (1% O2) or grown as xenografts in mice. The analyzed effects of EVO are compared to those of temozolomide (TMZ) and include HIF-1alpha protein expression, ATP and lactate production, cell viability and apoptosis (in cell cultures) as well as tumor size, survival and metabolite production in xenografted mice. The authors conclude that EVO acts by inhibiting glycolysis as a result of inhibition of HIF-1alpha protein expression. Overall this study is descriptive and does not provide any new or interesting mechanistic information. Furthermore, that are several technical shortcomings as described below.
Figure 1B. The authors should provide representative immunohistochemistry images of HIF-1a/PDK1 “high” and HIF-1a/PDK1 “low” samples.
Figure 2A. The subcellular fractionation appears to be inefficient as significant amounts of the cytosolic marker beta-actin appear in the “nuclear” fraction and the majority of the nuclear marker PCNA appears in the “cytosol” fraction. In the methods the authors mention that tubulin and lamins have been used as markers, but these do not appear in the figure. The authors should provide better fractionation results, should add analysis of total cell extracts corresponding to the fractions and should run both normoxic and hypoxic samples on the same gel/blot in order to allow direct comparison and document HIF-1a induction under hypoxia. They should also mention the cell line which the blots in Fig. 2A correspond to.
Fig. 4A. First of all, the blot images on the left appear to be composite images from separate individual lanes. The authors should analyze all samples (normoxic and hypoxic) with the same treatment or the same cell line on the same gel/blot and provide new images. Second, the results contradict Fig. 2A, as G06A cells appear not to express HIF-1a under normoxia in Fig. 4A while Fig. 2A shows similar HIF-1a expression between all three cell lines. Third, as hypoxic cells treated with EVO have high rate of apoptosis, the observed downregulation of HIF-1a protein expression may not be a specific effect but may reflect the downregulation of vital processes, such as transcription and translation, in the dying apoptotic cells under the cytotoxic effect of EVO. To reach a definite conclusion concerning the effect of EVO on HIF-1a expression, the authors should demonstrate that these processes are still functional at basal levels, should determine the levels of HIF-1a mRNA expression and should also check the protein expression of another short-lived protein.
Figs. 4B and 4C. The results of untreated cells should also be included as control.
General: Based on metabolite analysis, the authors connect the EVO-mediated downregulation of HIF-1a to inhibition of glycolysis. However, they provide no evidence for a direct causative connection. They authors should include data showing the levels of expression of known HIF-1 target genes regulating metabolic pathways under their experimental conditions, both in vitro and in vivo. They should also provide additional data concerning the mechanism through which EVO inhibits HIF-1a protein expression, if this is indeed a specific effect and not the result of shutting down global protein expression due to the cytotoxic effect of EVO (see also comment above).
Minor comments: Abbreviated terms such as Br-IMP, TMZ, CCNU etc should be explained.
Comments on the Quality of English LanguageOnly minor corrections are needed.